# Brief Communication: Updated GAMDAM Glacier Inventory over High Mountain Asia

Akiko Sakai

5 Graduate School of Environmental Studies, Nagoya University, Nagoya, Japan

*Correspondence to*: Akiko Sakai (shakai@nagoya-u.jp)

**Abstract.** The original Glacier Area Mapping for Discharge from the Asian Mountains (GAMDAM) glacier inventory was the first methodologically consistent dataset for High Mountain Asia. Nonetheless, the GAMDAM inventory underestimated glacier area as it did not include steep ice- and snow-covered slopes or shaded components. During revision of the inventory, Landsat imagery free of shadow, cloud, and seasonal snow cover was selected for the period 1990–2010, after which > 90% of the glacier area was delineated. The updated GAMDAM inventory, comprising 453 Landsat images, includes 134,770 glaciers with a total area of $100,693 \pm 11,790$ km$^2$.

## 1      Introduction

Glaciers in High Mountain Asia (HMA) play a significant role as a water resource for people living downstream (Immerzeel et al., 2010; Bolch et al., 2012). Glacier recession in recent decades has contributed to sea level rise, and this trend is anticipated to continue in the future (Huss and Hock., 2015; Marzeion et al., 2018; Radić and Hock, 2013). Recent analysis of surface elevation change has revealed that glaciers in HMA exhibit contrasting behavior (Brun et al., 2017; Gardner et al., 2013; Kääb et al., 2012, 2015): those in the Himalaya and East Nyainqentanglha Mountains are shrinking rapidly, while Karakoram and West Kunlun glaciers are undergoing are in balance or show a slight mass gain. Accordingly, a recent climate analysis for those areas demonstrated that the Karakoram and West Kunlun regions are relatively stable under global warming conditions, being less sensitive to temperature change (Sakai and Fujita, 2017). This assessment of both glacier volume and climatic conditions is based on a large-scale glacier inventory, highlighting the need for accurate, high-quality coverage of the entire HMA region. Specifically, precise glacier inventories are needed for modeling total glacier volume (Frey et al., 2014; Farinotti et al., 2019), deriving volume change from altimetry and DEMs (e.g., Brun et al., 2017) and surface-flow velocity (Dehecq et al., 2019), establishing changes in snow cover and albedo (Naegeli et al., 2019), catchment- and regional-scale hydrologic modeling (e.g., Immerzeel et al., 2010), projecting future glacier configuration (Huss and Hock, 2015; Shannon et al., 2019), and assessing uncertainty in estimates of glacier-surface elevation change (e.g., Nuimura et al., 2012; Bolch et al., 2017).

While the Randolph Glacier Inventory (RGI) (Arendt et al., 2015; RGI Consortium, 2017) was the first database with global coverage, the record exhibits considerable variability in accuracy even within HMA. Regional databases include the second Chinese glacier inventory (hereafter the CGI2), produced by automatic delineation with manual correction (Guo et al., 2015), and the NM18 inventory for the Karakoram and Pamir region (Mölg et al., 2018), derived from automated digital mapping and corrected manually by the coherence of synthetic aperture radar (SAR) imagery for debris-covered glaciers (Frey et al., 2012). The latter study also made separate delineations for all debris-covered areas.

Between February 2011 and March 2014, the Glacier Area Mapping for Discharge from the Asian Mountains (GAMDAM) project compiled a glacier inventory for HMA, covering the region in 27.0°–54.9°N latitude and in 67.4°–103.9°E longitude.

In its first iteration, published in 2015, the GAMDAM Glacier Inventory (GGI) did not include steep ice- and snow-covered slopes. Moreover, where wintertime imagery was employed to avoid summer monsoon cloud cover, shaded areas of glacier surfaces were excluded from the inventory (Fig. S1a). To help address these shortcomings, I present a revised glacier inventory for HMA based on summertime (May–September) imagery exhibiting clear glacier boundaries for steep, snow-

covered slopes and shaded areas. The abbreviated terms GGI15 and GGI18 refer to the first version of the GGI (Nuimura et al., 2015) and the current, updated version (this study), respectively.

## 2      Data

I utilized a total of 453 Landsat 5 Thematic Mapper™ and Landsat 7 Enhanced Thematic Mapper (ETM+) Level 1T scenes derived from 196 USGS EarthExplorer path-row sets (http://earthexplorer.usgs.gov/). Landsat ID and acquisition dates were

used to delineate glacier outlines and are summarized in Table S1. Due to the challenge of obtaining summertime imagery for the 1999–2003 setting period (Nuimura et al., 2015) that is free of clouds, seasonal snow cover, and shadows, the annual search range was expanded to 1990–2010 and the monthly search range to May–September (i.e., the high-solar-angle season). Where part of a glacier surface was obscured by cloud or snow, the Landsat archive was searched for more viable images covering that particular site; for glaciers with steep headwalls, images were selected with the most clearly defined

glacier outlines (full details of this methodologic approach are given in Section 3). As a result, the GGI18, like its predecessor, contains single path-row scenes comprising multiple images (Fig. 1). Finally, the GGI18 employs the ASTER-GDEM2 to analyze the glacier aspect in each $90 \times 90$ m grid.

## 3      Methods

Unlike seasonal snow cover, glaciers are considered to be permanent snow and ice. It is vital, therefore, that seasonal snow

coverage is excluded from each glacier polygon. In addition, to help quantify the glacial contribution to sea-level change and water resources, polygons must include all areas in which fluctuations in surface elevation reflect changes in ice mass.

### 3.1      Selection of Landsat imagery

As detailed in Section 2, I expanded the search period to obtain Landsat images in which glacier outlines are depicted clearly.

Figure S1, for example, shows the five images selected to delineate glacier outlines in the accumulation zone of the Khumbu Glacier, Nepal Himalaya. While the cloud-free image in Figure S1a was utilized for the GGI15, large areas of the glacier surface lie in shadow, thus precluding accurate delineation. Therefore, during revision for the GGI18, I selected an additional four images (Fig. S1b and c) with minimal snow and no cloud cover over the target glaciers (yellow squares in left panels, Fig. S1). Focusing on the steep snow-covered headwalls of the Khumbu Glacier (purple ellipses in right panels, Fig. S1), the

image displayed in Figure S1b exhibits the least seasonal snow cover and provides the sharpest boundaries among the four additional images, and thus was utilized in the GGI18.

Ultimately, the degrees of cloud and snow cover, and the clarity of glacier outlines, are the key factors in selecting suitable Landsat imagery for glacier delineation. The most challenging sites are those for which the glacier headwall comprises at least part of the accumulation area; to delineate such glaciers accurately, I focused on unambiguous boundaries on north-facing walls. Nonetheless, in regions dominated by summer monsoonal precipitation, such as the Eastern Himalaya and East Nyainqentanglha Mountains, the approach described here was inadequate to locate appropriate imagery (see Section 4.3).

## 3.2 Manual delineation

Owing to the many debris-covered glaciers in HMA (e.g., Herreid et al., 2015; Minora et al., 2016; Nagai et al., 2016; Ojha et al., 2017), for which automatic detection using the band ratio method is not possible (Paul et al., 2002), all glacier outlines included in the GGI18 were delineated manually. Using the newly selected Landsat imagery, I modified the GGI15 glacier polygons following the method described by Nuimura et al. (2015), but with two important differences. First, whereas glaciers of $< 0.05$ km$^2$ in area were excluded from the GGI15 (Nuimura et al., 2015), the minimum glacier area in the GGI18 is $0.01$ km$^2$ so as to account for the numerous small glaciers separated by dividing ridges. Furthermore, I included small glaciers as much as possible during the revision process. 10 grid cells (= $0.009$ km$^2$) were used as a guide for measuring area. In contrast to the GGI15, in which glacier outlines were delineated manually by 11 individuals (Nuimura et al., 2015), all of the delineation for the GGI18 was conducted by a single person.

The second methodological difference between the GGI15 and the GGI18 relates to steep headwalls. Nuimura et al. (2015) excluded steep snow- and ice-covered slopes from the GGI15, arguing that glaciers on high-angle headwalls generally do not undergo changes in surface elevation related to mass fluctuations. Those authors also underestimated the scale of upper glacier headwalls that are mantled with snow or ice. In contrast, since I was able to obtain comparatively distinct outlines for those glaciers with relatively thick ice on steep headwalls, the GGI18 includes all snow- or ice-covered parts of the glacier surface. For instance, Figure S2a depicts the high-angle, avalanche-prone headwall of the Trakaring Glacier in 2016, on which hanging glaciers are clearly visible. Thanks to their distinct outlines, these features are also identifiable on the 1999 Landsat image (arrows, Fig. S2b), indicating that they are long-term components of the glacier system and thus need to be included in the inventory.

Rock glaciers which develop independently from debris-covered glaciers can be identify since they have no snow accumulation area and have typical wrinkled landform. While, rock glaciers typically develop at the termini of debris-covered glaciers at the Karakoram and the Himalaya. They can be distinguished from debris-covered glaciers, which have

ponds surrounded ice cliffs, because they have ridges and furrow surface patterns (Bodin et al. 2010). Then, debris-covered areas were determined from high-resolution Google Earth imagery. Specifically, those portions of the glacier surface exhibiting rock glacier-like topography (e.g., flow lobes), were identified visually and omitted (see Fig. S3). As for the debris-covered glaciers in the Eastern Himalaya and East Nyainqentanglha Mountains, crevassed surface can be detected even in the snow covered glacier surface using high-resolution Google Earth imagery. For regions where high-resolution Google Earth imagery is unavailable (e.g., Eastern Himalaya and East Nyainqentanglha Mountains) or the glacier surface is obscured by seasonal snow cover (e.g., Karakoram and Pamir), I employed a combination of contours and surface-color difference between glacier and glacier-free areas to delineate debris-covered glaciers.

## 3.3    Uncertainties in glacier area

Revision of glacier outlines and subsequent delineation testing were both performed by the author. Delineation tests were conducted on 10 debris-covered glaciers and 12 debris-free glaciers using a total of 10 Landsat images (listed in Table S4), which included shaded (winter), snow-covered, and partially cloud-covered scenes. Since fully cloud-obscured images were not used in the delineation process, I did not select such glacier outlines in the testing process. Furthermore, I did not utilize Google Earth imagery since the resolution is not regionally uniform throughout HMA (see Section 3.2). For each Landsat image, I created a single glacier outline and calculated the normalized standard deviation (NSD: standard deviation divided by average glacier area) for each glacier area (e.g., Fig. S4). For each area class, the NSD increases with decreasing glacier area (Fig. S5). Moreover, NSD values are higher for debris-covered glaciers than for debris-free glaciers (particularly for smaller glaciers), although the GGI18 does not classify debris-covered and debris-free glaciers.

The proportion of debris-covered glaciers in each area class in the Eastern Himalayas (85.0°–92.0°E, 27.5°–29.0°N) (Ojha et al., 2017) (Fig. S6) was applied for all study area (HMA), then, they are used to calculate the number-weighted average NSD of glacier area for each glacier area class, including both debris and debris-free glaciers (Fig. S6). Here, the NSDs of the glacier area were assumed to be 15% for smaller ($< 0.25$ km$^2$) debris-free glaciers and 30% for smaller ($< 2$ km$^2$) debris-covered glaciers based on Fig. S5. NSD for all glaciers in Fig. S6 was assumed to be the uncertainty in glacier area for all types of glacier (including debris-covered and debris-free). Finally, the maximum NSD 19% was found for glaciers with 1-2 km$^2$ in area (Fig. S6).

## 4    Results and discussion

The GGI15 reported a total glacier area of 91,263 ± 13,689 km$^2$ (Nuimura et al., 2015), which included the combined area of holes in glacier polygons. Excluding polygon holes, I recalculated the total glacier area in GGI15 as 87,583 ± 3137 km$^2$ (Table 1).While, the GGI18 comprises 134,770 glaciers with a total area of 100,693 ± 11,790 km$^2$ (Table 1). Therefore, the total glacier area and glacier number for HMA are 13% and 35% greater, respectively, in the GGI18 than in the GGI15.

### 4.1    Comparison with the GGI15

Following the region delimitation of RGI 6.0 (Arendt et al., 2015; RGI Consortium, 2017), the aggregated polygon files for the GGI18 are divided into four regions: Central Asia, South Asia East, South Asia West, and North Asia (limited by the Sayan and Altai Mountains). Regional differences in glacier area among the GGI18, GGI15, and RGI 6.0 are summarized in

Table S2 (note that the RGI 6.0 incorporated part of the GGI15; RGI Consortium [2017]). For all regions, glacier area in the GGI18 is > 10% greater than in the GGI15, with the greatest differences in South Asia East (+18%) and South Asia West (+16%). Both South Asia East and West cover portions of the High Himalaya, including abundant high-relief glaciated headwalls, indicating that the GGI15 underestimated glacier area most in shaded areas. In the present study, I replaced glacier outlines delineated from winter imagery (GGI15) with those based on summer imagery (GGI18), with the result that

glacier area ratios based on summer images increased from 69% to 95% (Table 1). Figure 2 provides a comparison of a glacier outline included in both the GGI15 and GGI18 inventories. In the former, glacier delineation was based on low-solar-angle, heavily shaded imagery; in the latter, such areas have been substituted with delineations based on high-solar-angle imagery (Fig. S7).

Total glacier area in the GGI18 includes components on north-facing slopes (Fig. S8). However, the acquisition dates of the imagery are variable. For instance, the glacier area ratio derived from images acquired between 1999 and 2001 decreased from 73% in the GGI15 to 48% in the GGI18 (Table 1). For both inventories, glacier area distributions for specific acquisition dates (month and year) are compared and summarized in Figure S9. Glaciers located in monsoon-dominated regions were delineated primarily from non-summer (January–May and October–December) imagery in the GGI15 (Fig. S9a

and b), whereas the majority of the total glacier area (> 90%: Table 1) was extracted from summer (June–September) Landsat imagery (Fig. S9c).

According to the area-elevation distributions shown in Figure S10a, total glacier area between 5000 and 6000 m elevation is greater in the GGI18 than in the GGI15. While glacier area in the GGI18 is measurably larger across all area classes (Fig.

S10c), the greatest increase in glacier number is observed for small (< 0.0625 km$^2$) glaciers (Fig. S10b). Glacier polygons were aggregated for each 1° × 1° grid based on the barycentre of each glacier polygon for each inventory, to assess regional differences (see Fig. S10d). Compared with the GGI15, the GGI18 exhibits higher glacier-area values in all regions except the Tibetan Plateau (Fig. S10d), where the general absence of high-relief terrain minimizes the magnitude of topographic shading.


### 4.2    Comparison with the CGI2 and NM18 inventories

To assess the GGI18 relative to the CGI2 (Guo et al., 2015) and NM18 (Mölg et al., 2018) inventories, I extracted the two components of the GGI18 covered by the respective domains of the other datasets. A direct comparison of the three reveals that the GGI18-derived glacier area is smaller than that of the CG12 for elevations of 4000–5500 m (Fig. S11a) and lower

than that of the NM18-derived estimate for elevations of 4500–6000 m (Fig. S12a). In contrast, the GGI18 reports a greater number of smaller glaciers than do the CG12 and the NM18, ~~although~~and larger glaciers comprise a smaller total area in the GGI18 (Figs. S11b, c and S12b, c). This pattern is likely due to the greater division in the GGI18 of large ice masses into multiple glaciers relative to the NM18 and CGI2.

For each $1° \times 1°$ grid cell, glacier polygons for all three inventories were aggregated based on the polygon barycentre, thereby enabling regional differences to be calculated (Figs S11d and S12d). According to this comparison, glacier areas provided by the GGI18 and CG12 are regionally consistent (Fig. S11d), with the exception of the Nyainqentanglha Mountains, for which the CGI2 was not updated following the first Chinese glacier inventory. In contrast, compared with the NM18, the GGI18 prescribes a slightly smaller glacier area for most regions (Fig. S12d). This disparity is potentially linked to the inclusion of seasonal snow in the NM18, due to the automatic band-ratio method employed over debris-free zones (Mölg et al., 2018), whereas the GGI18 tends to omit such small glaciers. Furthermore, during revision for the GGI18, delineation of debris-covered glacier termini in the Karakoram and Pamir was hampered by seasonal snow cover in the high-resolution Google Earth imagery. Finally, I evaluated the degree of consistency between the GGI18 and the other two inventories using an overlapping ratio. This assessment provided an overlapping ratio of 87% for the GGI18 and NM18, and a ratio of 86% for the GGI18 and CGI2 to the total GGI18 over their respective domains (NM18/CGI2) (Table S3), indicating a high degree of consistency among the three inventories.

### 4.3     Glacier outlines requiring further revision

Clouds, seasonal snow cover, and strong shadows all hamper the detection of glacier outlines from Landsat imagery. Consequently, the number of scenes required to delineate glacier outlines for each path-row varies widely (Fig. 1), with monsoon-dominated regions utilizing the most imagery. Example of glacier outlines within such a limited area delineated using multiple images were shown in Fig S13. Therefore, the number of images in Figure 1 represents the degree of delineation accuracy.

25

As cloud-free and least seasonal snow satellite imagery becomes available, from existing sources other than Landsat or in the future, the glacier outlines delineated here from multiple images need to be revisited and, if necessary, revised. Sentinel-2 imagery, for instance, might prove a suitable alternative owing to its high resolution and shorter acquisition interval (≤5 days) relative to Landsat.

30

### 5     Summary

The updated version of the GAMDAM glacier inventory, the GGI18, incorporates all of HMA and includes 134,770 glaciers covering $100,693 \pm 11,790$ km$^2$. Although nearly 95% of the total glacier area was delineated from summer images, the acquisition date of source imagery varies widely. Relative to its predecessor (GGI15), the total glaciated area in HMA is

~15% greater in the GGI18, due primarily to the inclusion of glaciated north-facing slopes. Owing to cloud, seasonal snow cover, and topographic shading, a number of path-row scenes required multiple Landsat images to delineate glacier outlines fully and thus should be revisited in the future as higher-quality imagery becomes available.

*Data availability.*

5 (Data submitted to PANGEA are under review. Titles and abstracts can be revised after the review but are currently protected by password.)

1. Sakai, A. (2018): GAMDAM glacier inventory for High Mountain Asia: Area–altitude distribution for Central Asia. https://doi.pangaea.de/10.1594/PANGAEA.891415

2. Sakai, A. (2018): GAMDAM glacier inventory for High Mountain Asia: Area–altitude distribution for North Asia. 10 https://doi.pangaea.de/10.1594/PANGAEA.891416

3. Sakai, A. (2018): GAMDAM glacier inventory for High Mountain Asia: Area–altitude distribution for South Asia East. https://doi.pangaea.de/10.1594/PANGAEA.891417

4. Sakai, A. (2018): GAMDAM glacier inventory for High Mountain Asia: Area–altitude distribution for South Asia West. https://doi.pangaea.de/10.1594/PANGAEA.891418

15 5. Sakai, A. (2018): GAMDAM glacier inventory for High Mountain Asia: Central Asia in ArcGIS (shapefile) format. https://doi.pangaea.de/10.1594/PANGAEA.891419

6. Sakai, A. (2018): GAMDAM glacier inventory for High Mountain Asia: North Asia in ArcGIS (shapefile) format. https://doi.pangaea.de/10.1594/PANGAEA.891420

7. Sakai, A. (2018): GAMDAM glacier inventory for High Mountain Asia: South Asia East in ArcGIS (shapefile) format. 20 https://doi.pangaea.de/10.1594/PANGAEA.891421

8. Sakai, A. (2018): GAMDAM glacier inventory for High Mountain Asia: South Asia West in ArcGIS (shapefile) format. https://doi.pangaea.de/10.1594/PANGAEA.891422

*Competing interests.* The author declares no conflicts of interest.

*Acknowledgements.* This project was supported by a grant from the Funding Program for Next Generation World-Leading Researchers (NEXT Program, GR052) and Grants-in-Aid for Scientific Research (26257202) of the Japan Society for the Promotion of Science. I wish to thank all members of the GAMDAM project for their valuable support in producing the first
version of the GAMDAM Glacier Inventory.

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

**Table 1.** Comparison of the GGI15 and GGI18 inventories in terms of the total glacier area, glacier area ratio based on summer imagery, and that based on imagery acquired between 1999 and 2001.

| | Minimum glacier area (km$^2$) | Total glacier area (km$^2$) | Total number of glaciers | Number of Landsat images employed | Glacier area based on summer (JJAS) images (%) | Glacier area based on images acquired from 1999 to 2001 (%) |
|---|---|---|---|---|---|---|
| GGI15 Nuimura et al. (2015) | 0.05 | 87,583 ± 13,137 | 87,084 | 356 | 69 | 73 |
| GGI18 This study | 0.01 | 100,693 ± 11,790 | 134,770 | 453 | 95 | 48 |

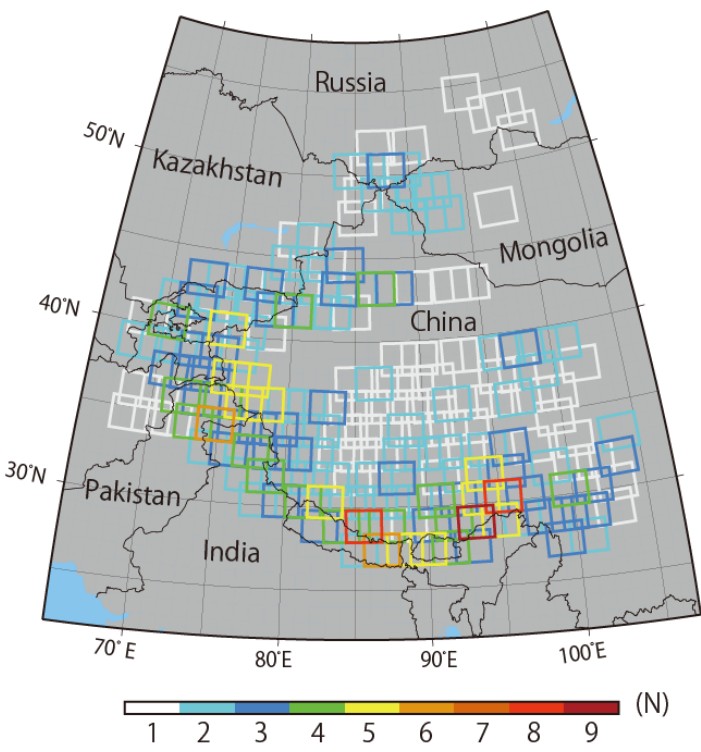

**Figure 1:** Footprints of the Landsat scenes used in the GGI18. Colors indicate the number of scenes used to delineate glacier outlines.

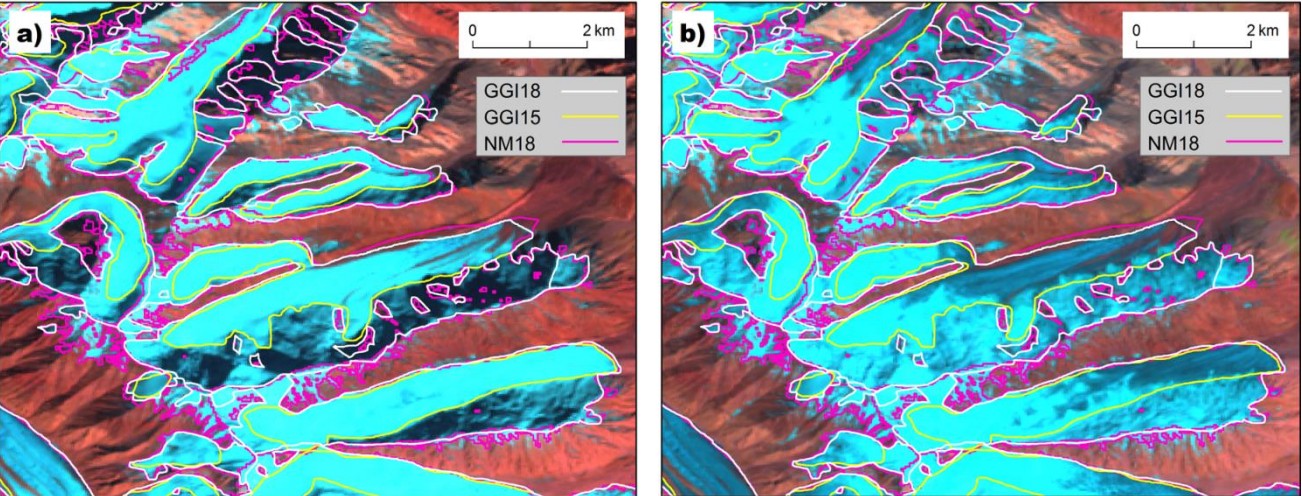

**Figure 2:** Comparison of glacier outlines used in the GGI15, GGI18, and NM18 inventories at 72°25'18"E, 38°55'25"N (path151 row33 of WRS2). Backgrounds are false-color (bands 7, 4, and 2 as RGB) composite Landsat images taken on 28 September 2001 (a) and 26 July 2001(b). Glacier outlines of the GGI15 (yellow lines) were delineated based on the strongly shaded image at left, whereas those of the GGI18 (white lines) were delineated using the less-shaded image at right. Glacier outlines of the NM18 (Mölg et al., 2018) (pink lines) are also shown for comparison.