# Peer review of "Brief Communication: Updated GAMDAM Glacier Inventory over High Mountain Asia"

_The Cryosphere, 2018_

## Referee Comment (RC1) · W.Q. Guo (Referee) · 21 Aug 2018

**Comments on "Brief Communication: Updated GAMDAM Glacier Inventory over the High Mountain Asia" submitted to *The Cryosphere Discussions* by Akiko Sakai**

Wanqin Guo, August 2018

**General Comments**

This manuscript reports the new updated GAMDAM Glacier Inventory over the High Mountain Asia. The author has done great and respectable jobs on manually revise the previous versions of GAMDAM Glacier Inventory only by herself, which must involves tremendous work loads. As a manuscript for brief communication, this paper is generally well writen. But from my point of view, some aspects should be revised to further improve the paper quality. I can understand that as a brief communication, the manuscript should be as brief as possible. But the author should describe some details on the methods or processes of the data revision (see suggestions in **Specific Comments**), or at least in the supplementary document. The absence of suck details makes current version very ambiguous thus hard to follow.

**Specific Comments**

**Page 2:**

Line 14-16: The first half of this sentence is somehow repititive with previous one ("… are relatively stable"). It's better to be rewrite properly.

**Page 3:**

Line 4: There's no other authors in this manuscript. Should you use "I" instead? Or is this refer to GGI15 data sources? It should be clearly marked if so.

Line 7: "seasonal cloud-free, seasonal-snow-free", seasonal in this part seems repititive, maybe remove the second "seasonal-" will be better. Besides, add "incidence" or other confining words between "solar angle" may be better.

Line 25-27: Although it's not consistent with common sense glacier inventory, Nuimura *et al.* (2015) provides a meaningful reason to exclude steep wall beyond the glaicer. Same to that paper, you should also give a reason that why you choose to include the snow- and ice-covered steep wall into the glacier, or researchers' comments or suggestions on this. You should better also mention the criteria of the steep wall here, e.g. same as the definitions in Nuimura et al. (2015) (>40°)? Or just by visual judgements?

Besides, the criteria you used to judge a patch as perennial snow or ice is also very important and should better to be presented somewhere, e.g. just as you saw in primarily used Landsat images acquired during one of the ablation season? Or carefully validated through comparing multiple images? Normally these steep back walls are always in rapid changes due to unpredictable snow/ice avalanches and/or frequent orographic snowfalls that may occur at anytime.

Line 27-28: It's not very clear from this sentence that which part of the debris-covered glacier were omitted. Maybe add a figure to illustrator the criteria will be much better.

**Page 4:**

Section 3.2: No methodology was described on how the quality of Landsat images was evaluated. It's a very important content for readers and potential glacier inventory compilers to understand the impacts of Landsat image quality on glacier inventory compilation.

Line 10-11: I suppose that one or a series criterion(a) was(were) used to evaluate the image

quality and assign the three quality ranks (A, B, and C) to each image. It will be too subjective if only using human judgements to do that work. So It's necessory to describe the method(s) on how the ranks were evaluated on a Landsat image, at least in the supplementary material.

Line 14: What is the score represented? And how the score was assigned to each factor on each image? It's fairly obscure in this section.

Line 26: On Fig.2, it is suggested to add coordinates on each sub-figures and thus will be much convenient for readers who want check the glacier outlines shown in the figure.

Line 29: On Fig.S2, same as the suggestion for Fig.2. Besides, it's not very clear here on the meaning of glacier area shown in Fig.S2. Are the glacier area shown counted by pixels numbers in glacier facing to different aspect ranges? Or simple the area of glacier whose average aspect belong to those aspect ranges?

**Page 5:**

Line 10: Actually there's no rule of "one glacier has one termius" neither from the earliest WGI handbook (Müller *et al.*, 1977), or from GLIMS tutorial (Raup and Khalsa, 2007&2010), or the guideline for glacier inventory compilation (Paul *et al.*, 2009). Just like what is said in Raup and Khalsa (2010), it isn't always easy to delineate ice divide for these glaciers by human judgement or even by common DEM analyze. Some glaciers like the diffluent glaciers actually do have two or more termini, and some hanging glaciers that on a long slope with similar orientations are also difficult to tell where the divides are. Actually the ice divides for these glaciers are changing that may be caused by even slight differences in the accumulation rate on different parts or changes in the flow velocity. Some ice caps even don't have apparent terminus but just occuping a flat mountain top. So it's not always necessary to divid these glacier into individual glaciers with single terminus.

Line 12: It's not very clear here that how the glacier size can determine the glacier number. It should be clearly clarified.

Line 17-18: It's also unclear that how the glaciers in $1\times1$ degree grids in Fig.S6d and S7d were grouped and how the discrepancies between GGI18 and CGI2 as well as NM18 were calculated in those grids (according to their label/centre points? Or cut by grid boundaries?). What the size (three levels) of the gridded points represents is also ambiguous (largest/mean area of all glaciers in the grid?). It needs to be clarified to avoid confusions.

Line 19: Should "Figs. 7b, c and 8b, c" be "Figs. S6b, c and S7b, c"?

Line 21: See above comments on Line 10. It is not always necessary to divide the diffluent glaciers especially the hanging glaciers into more individual glaciers.

**Page 6:**

Line 2-3: Same as comments on Page 4, Line 10-11, descriptions on the methods and criteria to evaluate Landsat image qualities are needed somewhere in the manuscript for better readers' understandings on how they are evaluated.

Line 5-6: The regions you called here as "Hengduan" in Fig. S8 are actually composed by East Himalaya Mountain and East NyenChen Tanglha Mountain. These regions are also dominated thus heavily influenced by monsoon called as India Monsoon or South Asia Monsoon through the river valleys of Yarlung Zangbo, leading to very poor satellite images that are always covered by snow or clouds. Please correct it.

---

## Referee Comment (RC2) · F. Paul (Referee) · 29 Oct 2018

**Review of tc-2018-139 by A. Sakai**

**General comments**

The Brief Communication by A. Sakai summarizes the main steps and results for creating a revised version of the GAMDAM glacier inventory. Before I forward my comments, I want to congratulate the author for this huge achievement. The effort and dedication to create this dataset is hard to imagine. Over the past two decades, I have likely digitized or corrected outlines from several ten thousand glaciers, but getting 130,000+ in just 3 years done is really impressive (this is more than 60% of the entire RGI). Despite the mentioned regional quality issues of the dataset I am pretty sure that GAMDAM2 will be a highly valuable baseline dataset for the years to come. But now to my general comments:

(1) This study is about glaciers so a publication in The Cryosphere makes sense. However, the paper is just a description of a dataset. There is no scientific advance or analysis included warranting publication in TC. This might be an editorial decision but in my opinion this study should be published in ESSD rather than TC.

(2) I agree with the comments forwarded by reviewer 1 (W. Guo) and will thus only partly or shortly repeat them here.

(3) The English wording/grammar is not good. Although it is mostly still possible to understand the text, I strongly recommend having the final version of the revised ms corrected by a native speaker before resubmission. I will not further comment on grammar issues.

(4) Methods: I suggest adding here a short section explaining what a glacier is in the context of this study, how this definition has been implemented practically, and what has been done when a strict application was not possible (have outlines then be transferred from a previous inventory?). For example, in the supplement one could show the time series of available images for a particular small region and describe why a specific scene has finally been selected to map a glacier (regarding snow, cloud, shadow conditions) or which scenes have been selected to get a complete outline. This would also be helpful advice for others creating a glacier inventory under difficult mapping conditions. The practical implementation should describe how seasonal snow has been distinguished from perennial snow and completely snow covered ice. At best, also this is illustrated with one or two examples of such conditions (in the supplemental material) to understand the related decisions and improve traceability.

(5) In the methods section I would insert a further section on uncertainty assessment. Just saying it is 15% as before is not convincing in my opinion. Uncertainties will likely be much smaller for larger glaciers so that it will be closer to 5-10% overall. As uncertainty scales with glacier size, one possibility is using a size-dependent empirical function as shown in Pfeffer et al. (2014) for RGI data. Instead of an empirical function one might also use the buffer method (with ±1/2 pixel) to determine a more realistic uncertainty (all ice divides should be removed beforehand). Of course, for debris-covered glaciers uncertainties might be higher but this can be commented on. The likely best method to determine an uncertainty value for this dataset would be independent multiple digitizing (at least three times) of several (say 10-20) glaciers of different size and with different challenges (debris, shadow, snow). The related standard deviation of the resulting relative area differences would be a good uncertainty measure for this datatset. Finally, it would also be possible to select a region with clean glaciers, map them automatically (e.g. with a band ratio) and use them as a reference for uncertainty assessment of the manual digitizing (see also doi: 10.1016/j.rse.2017.08.038).

6. I suggest moving several of the illustrations from the supplement into the main text and arrange them differently (i.e. more compact). Fig. 1 should be the current Fig. 1 plus Fig. S8 side-by-side (S8 is providing key information about dataset quality!). Fig. 2 should be the current Fig. 2 (please add a) and b)) but side-by-side to save some space. Fig. 3 should be Fig. S3 and S4 side-by-side. The a) panels of both figures can be included or remain in the supplement. Figure 4 should be the d) panels of Fig. S5 and S6, also side-by-side. The a) to c) panels of both figures and all other figures (S1, S2) and tables can remain in the supplement.

7. When comparing the outlines from GMADAM2 with NM18, I would describe the differences more precisely. As also visible in the current Fig. 2, NM18 seems to have included many regions with seasonal snow and is thus clearly overestimating glacier area and the number of small glaciers. As wrongly mapped seasonal snow has been mentioned in NM18 as a source of uncertainty, this can be confirmed here. I would also mention that there are sometimes larger differences in the extent of debris-covered glaciers between GAMDAM2 and NM18. In part, these might be due to the well-known difficulties in the interpretation or in-between glacier surges, but in comparison with very high-resolution GE images I have the impression that NM18 is often overestimating glacier extents, i.e. including parts that are actually rock glaciers. This might be due to the use of SAR coherence images in NM18 that might have included larger parts of them. By describing the observed differences more explicitly, the reader might also get a better impression of the main challenges and where special care has to be taken.

**Specific comments**
P2, L16: not sensitive to temperature change: I would write 'less sensitive'
P2, L18/19: I think the purpose of a consistent and precise glacier inventory for the region is less on relating glacier fluctuations (changes in length and area) to climate change (which is a very difficult task), but more to facilitate calculations that rely on exact glacier extents. This includes modelling of total glacier volume, spatially constrain calculation of elevation or volume changes from altimetry and DEM differencing or flow velocity and snow cover on glaciers, hydrologic modelling from the catchment to the regional scale, determination of future glacier extents and volume evolution, and by providing stable ground for uncertainty assessment. All these would be error prone without exact outlines.
P3, L20: How could you determine the size of a glacier (to decide that it is smaller than 0.01 km2) before its extent is digitised?
P3, L27: Please add some words on how many GE images have been consulted (can be very rough order of magnitude) and the percentage of images that supported the interpretation. We have not been very successful in finding suitable GE images for all glaciers in NM18.
P4, L18: For the total region, I would contrast directly here the main numbers for GGI18 with those for GGI15, including the percentage of change in number and area. In section 4.1 you can then describe specifics for the regional numbers (Table S2).

**Tables and Figures**
Please see point 6 of my general comments for general feedback on Figures.
P10, Table 1: I suggest transposing this table so that is has 7 columns and three rows. This would also help in keeping the ms compact.
P12, Fig. 2: Please use white instead of green lines, increase the brightness level somewhat and arrange both images side-by-side.
Tables S3 and S4: I suggest merging the two tables into one
Fig. S1: a) I suggest using yellow instead of green lines and white instead of red.

---

## Author Comment (AC1) · 27 Feb 2019

Reply to review comments by Prof. Wanqin Guo

Your comments are written by Century.
My reply are written by Arial and in Blue.
Revised part at main text and at supplement were written in red.

Thank you for your valuable and positive comments and suggestions. I apologize that I could not submit revised manuscript for a long time. I have revised my manuscript as your comments. My main revision was as follows.

1) I have decided that I remove Fig. S8 (Quality of Landsat imagery), because the classification of Landsat imagery were not objective. Further, instead of the Fig. S8, I can evaluate the quality of Landsat imageries by Fig. 1, which shows the number of Landsat imageries used to delineation for each path-row.
2) I have included revision of total area of GGI15, because area calculation of GGI15 by Nuimura et al.(2015) included holes in glacier polygons.
3) Revised manuscript has been substantially edited by native speaker. But, I have wrote in red for those portions which contents has changed.

Specific Comments

Page 2:
Line 14‐16: The first half of this sentence is somehow repititive with previous one ( "… are relatively stable"). It's better to be rewrite properly.
>> I have changed to 'a slight mass increase for former statements. (Page 2 line13)

Page 3:
Line 4: There's no other authors in this manuscript. Should you use "I" instead? Or is this refer to GGI15 data sources? It should be clearly marked if so.
>> Yes, I have to express 'I' instead of 'We'. I have checked all text, and revised them.

Line 7: "seasonal‐cloud‐free, seasonal‐snow‐free", seasonal in this part seems repititive, maybe remove the second "seasonal‐" will be better. Besides, add "incidence" or other confining words between "solar angle" may be better.
>> I think 'cloud‐free' doesn't need 'seasonal', then, I removed the former 'seasonal'. (Page 3 line 11)

Line 25‐27: Although it's not consistent with common sense glacier inventory, Nuimura et al. (2015) provides a meaningful reason to exclude steep wall beyond the glaicer. Same to that paper, you should also give a reason that why you choose to include the snow‐ and ice‐covered steep wall into the glacier, or researchers' comments or suggestions on this.

You should better also mention the criteria of the steep wall here, e.g. same as the definitions in Nuimura et al. (2015) (>40°)? Or just by visual judgements? Besides, the criteria you used to judge a patch as perennial snow or ice is also very important and should better to be presented somewhere, e.g. just as you saw in primarily used Landsat images acquired during one of the ablation season? Or carefully validated through comparing multiple images? Normally these steep back walls are always in rapid changes due to unpredictable snow/ice avalanches and/or frequent orographic snowfalls that may occur at anytime.

>>There are two main reasons for including glaciers at steep headwall. One is actual hanging glacier (glaciers at steep headwalls) has some thickness, then those thickness change should contribute glacier mass changes. Second reason is that I could found Landsat images with clear glacier boundary at steep walls by expanding searching period.

 As you wrote, we have excluded steep headwalls even where snow covered in the GGI15. Because steep headwalls generally do not experience changes in surface elevation related to glacier mass fluctuations. Further, we could not find images with clear glacier boundary at steep walls, because the initial setting searching period was only 5 years from 1999-2003.

But, I have changed my mind that when I have been to field observation. In the Fig. S2a), which I took at the Trakarding Glacier in the Nepal Himalaya, some glaciers at steep headwalls has more than several meters in thickness, then I thought those ice mass should contribute to total glacier ice mass. Further, I could found many images with clear boundary of relatively thick glacier outlines at steep headwalls, which make it possible to delineate glacier outlines at steep walls by expanding the period of the acquisition year from 1990-2010. I have added summary of the above statement in the '3.2 Manual delineation' (Page 4 line22-30) .

Line 27‐28: It's not very clear from this sentence that which part of the debris‐covered glacier were omitted. Maybe add a figure to illustrator the criteria will be much better.

>> I have added two samples in Fig. S3.

Page 4:

Section 3.2: No methodology was described on how the quality of Landsat images was evaluated. It's a very important content for readers and potential glacier inventory compilers to understand the impacts of Landsat image quality on glacier inventory compilation.

Line 10 - 11: I suppose that one or a series criterion(a) was(were) used to evaluate the image quality and assign the three quality ranks (A, B, and C) to each image. It will be too subjective if only using human judgements to do that work. So It's necessory to describe the method(s) on how the ranks were evaluated on a Landsat image, at least in the supplementary material.

Line 14: What is the score represented? And how the score was assigned to each factor on each image? It's fairly obscure in this section.

> Thank you for your above comments. It's right what you pointed out about quantifying the quality of Landsat images. I have decided that I remove this section, right colums of Table S1 (Quality of Landsat images; Cloud, seasonal snow, shadow) and Figure S8, because all evaluation of quality of Landsat images are subjective. Instead of the quality of LANDSAT images, the number of Landsat images used in the GGI18 (Fig. 1) can represents the difficulty of delineation of glacier outlines, in other words, quality of images. I also revised section '4.3 Glacier outlines required to revise by other satellite images' and the example of hard mapping area in Fig S13 (Page 7 line22-27).

Line 26: On Fig.2, it is suggested to add coordinates on each sub‐figures and thus will be much convenient for readers who want check the glacier outlines shown in the figure.

> I have already added the Longitude and Latitude in the explanation of the figure in the original manuscript.

Line 29: On Fig.S2, same as the suggestion for Fig.2. Besides, it's not very clear here on the meaning of glacier area shown in Fig.S2. Are the glacier area shown counted by pixels numbers in glacier facing to different aspect ranges? Or simple the area of glacier whose average aspect belong to those aspect ranges?

> I cannot add the coordinates in Fig. 2. because the each areas of GGI18 and GGI15 include whole region of HMA. Then, I added ' for whole HMA' in the legend.(Fig. S8 in the revised version)

> The aspect in Fig. S2 (Fig. S8 in the revised manuscript) was calculated based on pixel numbers in glacier polygons. I did not used average aspect of each glaciers. I added the method of calculation in the explanation of this figure. (Fig. S8)

Page 5:

Line 10: Actually there's no rule of "one glacier has one terminus" neither from the earliest WGI handbook (Müller et al., 1977), or from GLIMS tutorial (Raup and Khalsa, 2007&2010), or the guideline for glacier inventory compilation (Paul et al., 2009). Just like what is said in Raup and Khalsa (2010), it isn't always easy to delineate ice divide for these glaciers by human judgement or even by common DEM analyze. Some glaciers like the diffluent glaciers actually do have two or more termini, and some hanging glaciers that on a long slope with similar orientations are also difficult to tell where the divides are. Actually the ice divides for these glaciers are changing that may be caused by even slight differences in the accumulation rate on different parts or changes in the flow velocity. Some ice caps even don't have apparent terminus but just occuping a flat mountain top. So it's not always necessary to divide these glacier into individual glaciers with single terminus.

>I have misunderstood about the rule. As I wrote in the text, I didn't apply this rule to all glaciers. So, I have excluded these description.

Line 12: It's not very clear here that how the glacier size can determine the glacier number. It should be clearly clarified.

> I have excluded this sentence, because this sentence has been described following the previously mentioned statement.

Line 17‐18: It's also unclear that how the glaciers in 1×1 degree grids in Fig.S6d and S7d were grouped and how the discrepancies between GGI18 and CGI2 as well as NM18 were calculated in those grids (according to their label/centre points? Or cut by grid boundaries?).

>I have added ' For each 1° × 1° grid cell, glacier polygons for all three inventories were aggregated based on the polygon barycentre, thereby enabling regional differences to be calculated (Figs S11d and S12d).' in the text (Page 7 line4-5).

What the size (three levels) of the gridded points represents is also ambiguous (largest/mean area of all glaciers in the grid?). It needs to be clarified to avoid confusions.

>I have added ' The size of each circle indicates glacier area sum of GGI18 at each grid cell.' in the Figure explanation. And I thought 'Glacier area class in GGI18' in the Figure S6d might lead to misunderstand, then, revised the legend 'Glacier area of GGI2018' in the

revised version (Fig. S10d, S11d, S12d).

Line 19: Should "Figs. 7b, c and 8b, c" be "Figs. S6b, c and S7b, c"?
>>I have added several figures in supplyments. I have revised the figure number properly. (Fig. 7line1)

Line 21: See above comments on Line 10. It is not always necessary to divide the different glaciers especially the hanging glaciers into more individual glaciers.
>> This sentence doesn't relate to the rule 'one glacier has one terminus', which I miss-understood. Only compared the state of separation of glaciers in GGI18 and other inventories. So, I did not changed the content of this sentence. (Page 7 line1-2)

Page 6:
Line 2‐3: Same as comments on Page 4, Line 10‐11, descriptions on the methods and criteria to evaluate Landsat image qualities are needed somewhere in the manuscript for better readers' understandings on how they are evaluated.
>> I have removed the evaluation of Landsat image quality as I described above.

Line 5‐6: The regions you called here as "Hengduan" in Fig. S8 are actually composed by East Himalaya Mountain and East NyenChen Tanglha Mountain. These regions are also dominated thus heavily influenced by monsoon called as India Monsoon or South Asia Monsoon through the river valleys of Yarlung Zangbo, leading to very poor satellite images that are always covered by snow or clouds. Please correct it.
>> I have removed the evaluation of Landsat image quality as I described above. Then, ' Hengduan" has also removed. Instead, I described one example with figure, where it was very hard to delineate glacier outlines. I wrote the region name 'Eastern Himalaya'.

---

## Author Comment (AC2) · 27 Feb 2019

Reply to comments by Prof. Frank Paul

Your comments are written by Century.
My reply are written by Arial and in Blue.
Revised part at main text and at supplement were written in red.

Thank you for your valuable and positive comments and suggestions. I apologize that I could not submit revised manuscript for a long time. I have revised my manuscript as your comments. My main revision was as follows.

1) I have decided that I remove Fig. S8 (Quality of Landsat imagery), because the classification of Landsat imagery were not objective. Further, instead of the Fig. S8, I can evaluate the quality of Landsat imageries by Fig. 1, which shows the number of Landsat imageries used to delineation for each path-row.
2) I have included revision of total area of GGI15, because area calculation of GGI15 by Nuimura et al.(2015) included holes in glacier polygons.
3) Revised manuscript has been substantially edited by native speaker. But, I have wrote in red for those portions which contents has changed.

 (1) This study is about glaciers so a publication in The Cryosphere makes sense. However, the paper is just a description of a dataset. There is no scientific advance or analysis included warranting publication in TC. This might be an editorial decision but in my opinion this study should be published in ESSD rather than TC.
>> Thank you for your positive suggestion. I agree with your comments, but, when I received review reports, I have no time to respond to change the Journal. And paper of first version of GAMDAM Glacier inventory was published at TC, then I have selected same Journal.

(2) I agree with the comments forwarded by reviewer 1 (W. Guo) and will thus only partly or shortly repeat them here.
>> Please check my reply to comments by W. Guo.

(3) The English wording/grammar is not good. Although it is mostly still possible to understand the text, I strongly recommend having the final version of the revised ms corrected by a native speaker before resubmission. I will not further comment on grammar issues.

>> I have changed the company of English editing in this revised version. I hope the new company could edit English properly.

(4) Methods: I suggest adding here a short section explaining what a glacier is in the context of this study, how this definition has been implemented practically, and what has been done when a strict application was not possible (have outlines then be transferred from a previous inventory?). For example, in the supplement one could show the time series of available images for a particular small region and describe why a specific scene has finally been selected to map a glacier (regarding snow, cloud, shadow conditions) or which scenes have been selected to get a complete outline. This would also be helpful advice for others creating a glacier inventory under difficult mapping conditions. The practical implementation should describe how seasonal snow has been distinguished from perennial snow and completely snow covered ice. At best, also this is illustrated with one or two examples of such conditions (in the supplemental material) to understand the related decisions and improve traceability.

>> I have added one subsection '3.1 Selection of Landsat imagery' with explanation of image selection process and also added Figure S1 using five Landsat scenes. Fig. S1 includes selected scenes in GGI15 and four candidate scenes for GGI18. (Page3 line23-Page4 line8)

(5) In the methods section I would insert a further section on uncertainty assessment. Just saying it is 15% as before is not convincing in my opinion. Uncertainties will likely be much smaller for larger glaciers so that it will be closer to 5-10% overall. As uncertainty scales with glacier size, one possibility is using a size-dependent empirical function as shown in Pfeffer et al. (2014) for RGI data. Instead of an empirical function one might also use the buffer method (with ±1/2 pixel) to determine a more realistic uncertainty (all ice divides should be removed beforehand). Of course, for debris-covered glaciers uncertainties might be higher but this can be commented on. The likely best method to determine an uncertainty value for this dataset would be independent multiple digitizing (at least three times) of several (say 10-20) glaciers of different size and with different challenges (debris, shadow, snow). The related standard deviation of the resulting relative area differences would be a good uncertainty measure for this datatset. Finally, it would also be possible to select a region with clean glaciers, map them automatically (e.g. with a band ratio) and use them as a reference for uncertainty assessment of the manual digitizing (see also doi: 10.1016/j.rse.2017.08.038).2

>> Thank you for your comments with specific suggestions. In the revised manuscript, I

have carried out a delineation test for debris- and debris-free glaciers using 10 Landsat imageries (shadowed, snow-covered)(Fig. S4). And I obtained relations between mean glacier area and normalized standard deviations of glacier area (standard deviations of glacier area /mean glacier area) for debris-covered glacier and debris-free glaciers, respectively (Fig. S5). Because I did not classified debris- and non-debris-covered glaciers, ratio of debris-covered glacier's number at each area class at Eastern Himalayas (Ojha et al.(2017) ) were applied to estimate uncertainty of all glaciers including both debris and debris-free glaciers (Fig. S6). Then, I have assumed that the normalized standard deviations of glacier area were uncertainty of glacier area. Then, the average uncertainty of glacier area at whole study area become about 11% as shown in Table S2. These contents are written in section '3.3 Uncertainties in glacier area'.(Page 5 line 6-22)

6. I suggest moving several of the illustrations from the supplement into the main text and arrange them differently (i.e. more compact).
>> There is a rule that "Brief communications have a maximum of 3 figures and/or tables". Then, I cannot move figures in supplement to main text.

Fig. 1 should be the current Fig. 1 plus Fig. S8 side-by-side (S8 is providing key information about dataset quality!).
>> Thank you for your pointing out the importance of Fig. S8 (Quality of Landsat imagery) . But, as I wrote my reply to Guo, I have decided to remove the Fig. S8, because the quality of Landsat imagery was classified subjectively.

Fig. 2 should be the current Fig. 2 (please add a) and b)) but side-by-side to save some space.
>> I have added a) and b) and relocate them to be side by side.

Fig. 3 should be Fig. S3 and S4 side-by-side. The a) panels of both figures can be included or remain in the supplement. Figure 4 should be the d) panels of Fig. S5 and S6, also side-by-side. The a) to c) panels of both figures and all other figures (S1, S2) and tables can remain in the supplement.
>> There is a rule that "Brief communications have a maximum of 3 figures and/or tables". Then, I could not put figures to main text.

7. When comparing the outlines from GMADAM2 with NM18, I would describe the differences more precisely. As also visible in the current Fig. 2, NM18 seems to have

included many regions with seasonal snow and is thus clearly overestimating glacier area and the number of small glaciers. As wrongly mapped seasonal snow has been mentioned in NM18 as a source of uncertainty, this can be confirmed here. I would also mention that there are sometimes larger differences in the extent of debris-covered glaciers between GAMDAM2 and NM18. In part, these might be due to the well-known difficulties in the interpretation or in-between glacier surges, but in comparison with very high-resolution GE images I have the impression that NM18 is often overestimating glacier extents, i.e. including parts that are actually rock glaciers. This might be due to the use of SAR coherence images in NM18 that might have included larger parts of them. By describing the observed differences more explicitly, the reader might also get a better impression of the main challenges and where special care has to be taken.

>> Thank you for your positive comments. I have included the statements that 'NM18 might overestimate glacier extents because ...'. But, as you wrote, terminus of glaciers in Karakoram and Pamir regions are covered with seasonal snow in GE. So, it is hard to detect terminus location of debris-covered glacier in this region. I described about these problems in the text at Page 7 line 7-14.

Specific comments

P2, L16: not sensitive to temperature change: I would write 'less sensitive'

>> I have revised.(Page 2 line15)

P2, L18/19: I think the purpose of a consistent and precise glacier inventory for the region is less on relating glacier fluctuations (changes in length and area) to climate change (which is a very difficult task), but more to facilitate calculations that rely on exact glacier extents. This includes modelling of total glacier volume, spatially constrain calculation of elevation or volume changes from altimetry and DEM differencing or flow velocity and snow cover on glaciers, hydrologic modelling from the catchment to the regional scale, determination of future glacier extents and volume evolution, and by providing stable ground for uncertainty assessment. All these would be error prone without exact outlines.

>> I have added what you pointed out with references. (Page 2 line 17-22)

P3, L20: How could you determine the size of a glacier (to decide that it is smaller than 0.01 km2) before its extent is digitised?

>> The minimum glacier area : 0.01 $km^2$ has correspond with 10 grid cells (0.009 $km^2$) of

Landsat imageries. Then, I have included nearly 10 grids of glaciers, when I digitized. I have added the explanation in the text. (Page 4 line17)

P3, L27: Please add some words on how many GE images have been consulted (can be very rough order of magnitude) and the percentage of images that supported the interpretation. We have not been very successful in finding suitable GE images for all glaciers in NM18.
>> Yes, sometimes GE did not support to detect terminus of debris-covered glaciers in the Karakoram and Pamir because of the seasonal snow cover. And we have little high resolution Google Earth images in the East Nyen Chen Tanglha Mountain. But, I can not write the number of GE images are consulted to delineate glaciers (even roughly), because I have delineated glaciers individually, not based on the unit of Landsat imagery or Google Earth image. But, anyway I have wrote about the problem of Google Earth images in the section of 'Manual delineation'. (Page4 line 33- Page 5 line 3)

P4, L18: For the total region, I would contrast directly here the main numbers for GGI18 with those for GGI15, including the percentage of change in number and area. In section 4.1 you can then describe specifics for the regional numbers (Table S2).
>> I wrote contents on Table 1 in the head of section 4.(Page 5 line23-27) And in the section 4.1, I wrote on the regional difference between GGI15 and GGI18. (Page 5 line31- )

Tables and Figures
Please see point 6 of my general comments for general feedback on Figures.

P10, Table 1: I suggest transposing this table so that is has 7 columns and three rows. This would also help in keeping the ms compact.
>>I have transposed the table. (Page 12)

P12, Fig. 2: Please use white instead of green lines, increase the brightness level somewhat and arrange both images side-by-side.
>> I have revised.

Tables S3 and S4: I suggest merging the two tables into one
>> I have merged. (Fig. S9 in the revised version)

Fig. S1: a) I suggest using yellow instead of green lines and white instead of red.

>> I have changed the colors. (Fig. S7 in the revised version)

---

## Referee Report (RR1)

**Comments on the revised manuscript "Brief Communication: Updated GAMDAM Glacier Inventory over the High Mountain Asia" by Dr. Akiko Sakai**
Wanqin Guo, April 2019

**General Comments**

Firstly allow my appology on the later referee report.

I read carefully through the revised manuscript. The comments were generally well considered. However, the author added too many contents in response to some comments of Dr. Paul and I on previous version, and also many other contents, which make the manuscript not as "brief" as previous version. Since the author has deleted many aspects in this revised version, which largely simplified the questions aroused in previous version, many explanations are not necessary again (e.g. on the Landsat imagery selection, and delineation of glacier area on steep back-wall). See specific comments for some revise suggestions.

This revision aroused another question on the differences between rock glacier and debris-covered parts of glacier, which were not well discussed among researchers in glacier inventory compilation, and should be dealt with much more cautions. I suggest the author to look through the manuscript again and describe related contents well, but should also be in brief words.

**One suggestion on future works**

I am truly suspecting that the author's research group has overestimated the overall area uncertainty (15%) in both versions merely by comparisons between different delineation tests, which is much larger than the generally achievable areal accuracy (3-5%) by Landsat series suggested by many authors. I suggest Dr. Sakai or other interested researchers to do further works on precisely evalutating the areal uncertainties of both versions of GGI. It can be done by comparing the glacier outlines with those delineated from free high resolution images from Google Earth or BING map, etc. The direct comparisons suggested in Paul *et al.* (2013)@Annals of Glaciology or the definitive method used by us (in Guo *et al.*, 2015@Journal of Glaciology) are suggested, but also can be done by some other solutions. The comparisons should consider different circumstances, i.e. on typical debris-covered glaciers, and glaciers influenced by long lasting snow/cloud covers and also heavy cast shadow, as well as clean-ice glaciers with fine image quality, to provide an overview of the precise areal uncertainties achieved by GGI dataset.

**Some specific comments:**

**Page 2:**

Line 8: "millions of" should better to be "billions of" here considering the much large area of the HMA region in this manuscript.

Line 15: Suggest to delete "overall" after "less sensitive".

Line 17-22: This sentence seems too long and should better to be more summarized.

Line 26: Actually the CGI2 was compiled firstly by automatic glacier delineation, then by intensive and multi-round of manual corrections, although the manual works have completely changed the appearances of the glacier outlines.

Line 32: It's better to add "in" or "of" before "longitude" and "latitude".

**Page 3:**

Line 6: "final" may means that you will never change it in the future, but in Section 4.3 you mentioned this version may need further revison. Is it the right word?

Line 14-15: You may not having clearly defined glacier outlines when selecting the Landsat images.

Line 16: The citations to figures all through the manuscript are inconsistent (many Fig. and also many Figure).

Section 3.1: Since you have remove the section of Quality of Landsat images in previous version, it's not necessary to explain the selection of the imageries in such detail. Although the selection of Landsat image is really a hard work for you considering the vast spatial coverage of GGI dataset, it's a common challenge faced by all researchers want to accurately delineate glacier outlines using remote sensing methods, and all of them may use identical method you mentioned here. Therefore, I suggest to completely remove this section and related supplementary figure.

**Page 4:**

Line 12: "for which ……" seems not a common expression. Suggest to revise.

Line 16-19: "Furthermore, …… measuring area.", this part is not necessary and suggested to be deleted.

Line 21-29: I suppose that the author didn't use any criterion on the slope range when delineating glacier areas on steep back-wall, so just describe here on how to distinguish hanging ice/glacier from seasonal snow in prevous para is good enough (I think it should be done by visual check from multiple Landsat images). Therefore, this part is also not necessary and should better to be completely deleted.

Line 32-33: Not a question, but honestly to say, I cannot see the reasonability to exclude the lowest part as rock glacier in Figure S3c&d.

**Page 5:**

Section 3.3: This section can also be much shortened, by simply tell the method you used to evaluate the area uncertainty (multiple delineation test on different images, and expressed in NSD value, maybe in several sentences).

Line 31: It's better to add a blank space between "RGI" and "6.0". Same in other places.

Line 34: Should "RGI16" here be "RGI 6.0"?

**Page 6:**

Line 32: Should "Figs. 11b, c and 12b, c" be "Figure S11b, c and S12b, c"? And "greater" maybe ambugious here, suggest to use "more" or other word instead.

**Page 7:**

Line 19-22: "For instance, …… (Fig. S13c)", this part is also in too much detail, and suggested to be removed.

Line 25: Suggest to revise the last part of this sentence as "and shorter acquisition interval (≤5 days)".

Line 30: See comment on Line 6 in Page 3. Maybe "latest version" or "new version" is better.

---

## Author Response (AR2)

Reply to Editor (Tobias Bolch)

Thank you for informative comments. I wrote my reply in blue Arial font in this reply letter. And revised part in the text are also written in blue. Please note that the last revision in the text were remained in red.

The reviewer raised specifically two issues: The better description between rock glaciers and debris-covered glaciers. This is a valid point I am aware that this is quite difficult but I agree that some more details are needed. You may here also refer to Moelg et al. (2018).
> I have added some description on rock glaciers citing Bodin et al. 2010 (which was written in Moelg et al. 2018) at Page 4 line31-P5 line5.

The second issue is the uncertainty. I also believe that 15% is quite high. I agree to be conservative, but I cannot follow how you come up with the 15% and 30%. I am also missing results of the delineation tests in the main manuscript.
> I have tried to revise the latter part of the section 3.3 (Page 5 Line 18-).
 'The proportion of debris-covered glaciers in each area class in the Eastern Himalayas (85.0°–92.0°E, 27.5°–29.0°N) (Ojha et al., 2017) (Fig. S6) was applied for all study area (HMA), then, they are used to calculate the number-weighted average NSD of glacier area for each glacier area class, including both debris and debris-free glaciers (Fig. S6). Here, the NSDs of the glacier area were assumed to be 15% for smaller (< 0.25 km$^2$) debris-free glaciers and 30% for smaller (< 2 km$^2$) debris-covered glaciers based on Fig. S5. NSD for all glaciers in Fig. S6 was assumed to be the uncertainty in glacier area for all types of glacier (including debris-covered and debris-free). Finally, the maximum NSD 19% was found for glaciers with 1-2 km$^2$ in area (Fig. S6).'

I also want to mention that I do not agree with all of the reviewers comments (e.g. fully remove section 3.1 or L. 21-29 on page 4) and you also do not need to agree with all comments by the reviewer, but in case not please provide a good reason.
   > I hope I could give good reasons.

Please find my comments below:
P2L9: I think you can leave "millions" (see reviewers comments) as the glaciers are important but not really for all billion people living downstream.

> Thank you for your comments. I deleted the "millions".

P2L13: Considering the uncertainty it is not clear whether Karakoram glaciers are really gaining mass. I suggest to adjust to: "…are in balance or show a slight mass gain" or similar.

> I have revised as your suggestion.

L18: Update the Huss 2012 citation with the recent paper by Farinotti et al. 2019

> corrected

L22: Thanks for citing my work here. However, I did not consider the area uncertainty properly in my 2011 paper. Maybe you can refer to the 2017, TC paper or others which take the area uncertainty explicitly into account.

>Thank you for your comment. I have revised.

P3L2f: I do not understand what you mean. Can you please be more precise or add a figure in the supplement?

>I have added '(Fig. S1a)' at the end of the sentence.

P5L22ff (Results section): I like the structure better how it has been before. The second sentence now is already a comparison to GGI15 and needs to be presented there.

> I have changed the statement of the order as follows, 'The GGI15 reported a total glacier area of 91,263 ± 13,689 km$^2$ (Nuimura et al., 2015), which included the combined area of holes in glacier polygons. Excluding polygon holes, I recalculated the total glacier area in GGI15 as 87,583 ± 3137 km$^2$ (Table 1).While, the GGI18 comprises 134,770 glaciers with a total area of 100,693 ± 11,790 km$^2$ (Table 1).'

P7L7ff.: I do not fully agree with this statement. First of all Moelg et al. used also high resolution images in addition to the coherence images (see Fig. 3) and whether or whether the movement of rock glaciers are detected depends on their activity/flow velocity and the time difference between the SAR image acquisition.

> I see. I removed the statement.

L10f: What does complicated mean? Did Moelg et al. 2018 used similar images than you? If not then they must have had the similar problem.

>I revised to 'delineation of debris-covered glacier termini in the Karakoram and Pamir was

hampered by seasonal snow cover in the high-resolution Google Earth imagery.'

P7L25: What does clearer mean? More suitable images with better snow and cloud conditions?
> Yes, I have revised to 'cloud-free and least seasonal snow satellite imagery…'

One comment regarding the correct writing of "Nyen Chen Tanglha". You find also "Nyenchen Tanglha" or "Nyainqentanglha", You may write the alternative transcription in brackets the first time used.
> I have revised all "Nyen Chen Tanglha" to "Nyainqentanglha".

The layout of the supplement needs to be improved. Try to avoid figures without captions on one page and have consistent space between the figures and the captions.
E.g. Fig S1: There is a lot of uncaused space, maybe one example can be omitted and the caption shortened when a legend is included (e.g. for "with glacier outlines for the GGI15 and GGI18 inventories indicated by red and yellow lines, respectively"
>I removed two images. And deleted 'with glacier outlines for the GGI15 and GGI18 inventories indicated by red and yellow lines, respectively' in the caption.

Fig. S3: The single figures can be smaller so that 2 figures can be shown beside each other.
> corrected

Fig. S5: It is a bit large and should be cantered
> corrected

Reply to 2nd review by Dr. Guo

Thank you for careful check. I wrote my reply in blue Arial font in this reply letter. And revised part in the text are also written in blue. Revised part in the last time were remained in red in the text.

One suggestion on future works

I am truly suspecting that the author's research group has overestimated the overall area uncertainty (15%) in both versions merely by comparisons between different delineation tests, which is much larger than the generally achievable areal accuracy (3‐5%) by Landsat series suggested by many authors. I suggest Dr. Sakai or other interested researchers to do further works on precisely evalutating the areal uncertainties of both versions of GGI. It can be done by comparing the glacier outlines with those delineated from free high resolution images from Google Earth or BING map, etc. The direct comparisons suggested in Paul et al. (2013)@Annals of Glaciology or the definitive method used by us (in Guo et al., 2015@Journal of Glaciology) are suggested, but also can be done by some other solutions. The comparisons should consider different circumstances, i.e. on typical debris‐covered glaciers, and glaciers influenced by long lasting snow/cloud covers and also heavy cast shadow, as well as clean‐ice glaciers with fine image quality, to provide an overview of the precise areal uncertainties achieved by GGI dataset.

Thank you for your suggestion for my future work.
I will keep in mind to evaluate the uncertainties of those inventories.
As for the last comment about the error under different circumstances in GGI18, I have already evaluated the uncertainty. Please read section 4.3. You wrote that the detail of this section is not necessary in the below comments. But, I will not delete because the detail is the reply to your above comment.

Some specific comments:
Page 2:
Line 8: "millions of" should better to be "billions of" here considering the much large area of the HMA region in this manuscript.

> Here, "millions of" is underestimated expression, but "billions of" is overestimated, then, I only deleted "millions of" here.

Line 15: Suggest to delete "overall" after "less sensitive".

> deleted "overall"

Line 17‐22: This sentence seems too long and should better to be more summarized.

> I think this sentence is important to insist usefulness of glacier inventory. I have cited only on or two example. So, I did not remove.

Line 26: Actually the CGI2 was compiled firstly by automatic glacier delineation, then by intensive and multi‐round of manual corrections, although the manual works have completely changed the appearances of the glacier outlines.

> Thank you for the details on CGI2. I have revised to 'produced by automatic delineation with manual correction. '

Line 32: It's better to add "in" or "of" before "longitude" and "latitude".

> added "in"s.

Page 3:

Line 6: "final" may means that you will never change it in the future, but in Section 4.3 you mentioned this version may need further revison. Is it the right word?

> corrected 'updated version'.

   Actually, this is final version as the GAMDAM glacier inventory. But, the title of this paper is 'updated…'. For the consistency, I have revised 'updated version…' at line6 Page 3 and also line 30 in page 7.

Line 14‐15: You may not having clearly defined glacier outlines when selecting the Landsat images.

> I totally agree with you. I could not find clearly defined glacier outlines for all glaciers in HMA. Then, I include glacier outlines based on images with seasonal snow, shadow to evaluate uncertainty (see 3.3). Further, I wrote section 4.3 'Glacier outlines requiring further revision', which include seasonal snow cover.

Line 16: The citations to figures all through the manuscript are inconsistent (many Fig. and also many Figure).

> 'Figure' is used when it is at the head of the sentence. In other case ( in the middle or at the end of the sentences), we use 'Fig.'. Please check other papers.

Section 3.1: Since you have remove the section of Quality of Landsat images in previous version, it's not necessary to explain the selection of the imageries in such detail. Although the selection of Landsat image is really a hard work for you considering the vast spatial coverage of GGI dataset, it's a common challenge faced by all researchers want to accurately delineate glacier outlines using remote sensing methods, and all of them may use identical method you mentioned here. Therefore, I suggest to completely remove this section and related supplementary figure.

> Selection of Landsat image might be common challenge. But, if this section have to be removed, this paper is not valuable to publish, because change of source images is the main reason of the difference between GGI15 and GGI18. I have also discussed about the image selection at section 4.1 (Fig. S9).

Page 4:

Line 12: "for which ……" seems not a common expression. Suggest to revise.

> I think this word is not abnormal.

Line 16‐19: "Furthermore, …… measuring area.", this part is not necessary and suggested to be deleted.

> For line 16-17 Including small glaciers in GGI18 is main difference from GGI15. Then, I did not decide to remove sentences. And if I delete Line 18-19, there is no description that the revision is conducted by one (single) person. Then, I have not deleted the sentence.

Line 21‐29: I suppose that the author didn't use any criterion on the slope range when delineating glacier areas on steep back‐wall, so just describe here on how to distinguish hanging ice/glacier from seasonal snow in prevous para is good enough (I think it should be done by visual check from multiple Landsat images). Therefore, this part is also not necessary and should better to be completely deleted.

> Including steep head wall covered with ice in GGI18 is important difference. Further, there is no clear criterion on the slope range, then I have to describe the detail, here.

Line 32‐33: Not a question, but honestly to say, I cannot see the reasonability to exclude the lowest part as rock glacier in Figure S3c&d.

> The screen shot images of Google Earth might not show the detail, But, we can find winkles geometry in the excluded area.

Page 5:

Section 3.3: This section can also be much shortened, by simply tell the method you used to evaluate the area uncertainty (multiple delineation test on different images, and expressed in NSD value, maybe in several sentences).

> Thank you for your suggestion. But, if I remove the detail, readers will misunderstand the uncertainty. For example, only 'different images' indicates no information about the situation of images (e.g. with shadow or with seasonal snow…). And NSD have to be explained the calculation method. Then, I did not delete.

Line 31: It's better to add a blank space between "RGI" and "6.0". Same in other places.

> Added a space for all 'RGI 6.0'.

Line 34: Should "RGI16" here be "RGI 6.0"?

> corrected to "RGI 6.0"

Page 6:

Line 32: Should "Figs. 11b, c and 12b, c" be "Figure S11b, c and S12b, c"?

> corrected

And "greater" maybe ambugious here, suggest to use "more" or other word instead.

> Here, the sentence express 'greater number' specifically. I think 'more smaller glaciers…' is more ambugious.

Page 7:

Line 19 ‑ 22: "For instance, ⋯⋯ (Fig. S13c)" , this part is also in too much detail, and suggested to be removed.

> removed and added only one sentence.

Line 25: Suggest to revise the last part of this sentence as "and shorter acquisition interval (≤5 days)".

>I have added some words as your suggestion.

Line 30: See comment on Line 6 in Page 3. Maybe "latest version" or "new version" is better.

>For consistency, I have revised to updated version' as I wrote above reply.

---

## Author Response (AR3)

Reply to Editor (Tobias Bolch)

Thank you for your careful check.

I have corrected as your comments and corrected part in the text were written in red.

In this reply letter, your comments are written in Times New Roman.

And My reply are written in blue and Arial.

Page 2 line 13 "are undergoing are in " > I have removed 'are undergoing are in'

Page 4 line 25 "all" You might probably have missed some. Just write "the".

> I have corrected.

Page 4 line 31-You need to start differently as you do not map these talus-derived rock glaciers. State the problem and then how you solved it, e.g. A challenge is the correct distinction of debris-covered glacier and rock glaciers as gradual transitions can exists under permafrost conditions, as e.g. described for the Pamir (Moelg et al.)

I do not agree that the separation is easy, so please write the statement with more care

> I have revised as follows

'A challenge is the correct distinction of debris-covered glacier and rock glaciers as gradual transitions can exist under permafrost conditions. Rock glaciers have terrains with ridges and furrow surface patterns (Bodin et al. 2010), while debris-covered glaciers have ponds surrounded ice cliffs. Those detailed topography cannot be detected by Landsat imagery because of low resolutions.'

And also I have added following sentence at the end of the section. (Page 5 line 8)

'the method might lead to under- or over-estimate debris-covered glacier area.'

Page 5 line 31 Hence would be the better term.

> I have corrected.

Page 7 line9 "to" > I have corrected

Page 7 line12 "This contradicts what you wrote before about the distinguishing of rock glaciers and debris-covered glaciers." > I have removed this sentence.

---

## Author Response (AR4)

Dear Tobias,

Thank you for your careful check. I hope I could catch your intended meaning.

Your comments are written in Century. My replies are written in Blue and Arial in the reply letter. And revised parts are written in red in the text.

The sentence you added provides some explanation but it is still not clear. You cannot argue that "Those detailed topography cannot be detected by Landsat imagery because of low resolutions" and then write in the next sentence "Then, debris-covered areas were determined from high-resolution Google Earth imagery. Specifically, those portions of the glacier surface exhibiting rock glacier-like topography (e.g., flow lobes), were identified visually and omitted."

Please clarify; maybe write "Therefore ..." and remove the info in brackets as it is now written beforehand.

>I have revised as 'Those detailed topography were difficult to be detected by Landsat imagery because of low resolutions. Therefore,…'

Also improve "surrounded BY ice cliffs"

> I have added 'by'.

and add a reference for the sentence "as gradual transitions can exist under permafrost conditions" Maybe one of W. Haeberli or Moelg et al..

>I have added (Mölg et al., 2018) at line 32 in Page 4.

The added sentence "the method might lead to under- or over-estimate debris-covered glacier area." is not clear. It is too unspecific. Any estimate of the "under and over estimation"? Does this refere to the rock glacier problem?

> I have deleted the sentence.

---

## Author Response (AR5)

Dear Tobias,

Thank you so much for careful check of my manuscript.

I have corrected to "because of relatively low resolution." at the first line Page 5.

Thank you.

Best wishes,

Akiko